# Metagenomic Analysis Reveals the Anti-Inflammatory Properties of Mare Milk

**DOI:** 10.3390/ijms26178239

**Published:** 2025-08-25

**Authors:** Ran Wang, Wanlu Ren, Shibo Liu, Zexu Li, Luling Li, Shikun Ma, Xinkui Yao, Jun Meng, Yaqi Zeng, Jianwen Wang

**Affiliations:** 1College of Animal Science, Xinjiang Agricultural University, Urumqi 830052, China; 17590811761@163.com (R.W.); 13201295117@163.com (W.R.); 13898127437@139.com (S.L.); 13593312012@163.com (Z.L.); 18996888638@163.com (L.L.); 18299152719@163.com (S.M.); yxk61@126.com (X.Y.); junm86@sina.com (J.M.); xjauzengyaqi@163.com (Y.Z.); 2Xinjiang Key Laboratory of Equine Breeding and Exercise Physiology, Urumqi 830052, China

**Keywords:** mare milk, koumiss, anti-inflammatory properties, metagenomics

## Abstract

This study aimed to assess the anti-inflammatory properties of mare milk by analyzing immune markers in mice following gavage of mare milk. Metagenomic sequencing was employed to examine variations in the composition and functional profiles of the intestinal microbiota across different experimental groups. Bacterial diversity, abundance, and functional annotations of gut microbiota were evaluated for each group. The results show that, compared to the control group, the mare milk group exhibited a significant decrease in the pro-inflammatory cytokine IL-6 levels and a significant increase in secretory immunoglobulin A (SIgA) levels (*p* < 0.05). The fermented mare milk group and the pasteurized fermented mare milk group demonstrated a significant downregulation of the pro-inflammatory cytokines TNF-α and IL-1β, along with a significant increase in the anti-inflammatory cytokine IL-10 levels (*p* < 0.05). Additionally, metagenomic analysis revealed that both the mare milk and fermented mare milk groups were able to regulate the imbalance of the intestinal microenvironment by improving the diversity of the gut microbiota and reshaping its structure. Specifically, the mare milk group enhanced gut barrier function by increasing the abundance of Bacteroides acidifaciens, while the fermented mare milk group increased the proportion of Bacillota and the relative abundance of beneficial bacterial genera such as Faecalibaculum and Bifidobacterium. KEGG pathway annotation highlighted prominent functions related to carbohydrate and amino acid metabolism, followed by coenzyme and vitamin metabolism activities. In conclusion, mare milk and its fermented products demonstrate anti-inflammatory effects, particularly in modulating immune responses and inhibiting inflammatory cascades. Additionally, the administration of mare milk enhances the composition and metabolic activity of intestinal microbiota in mice, supporting intestinal microecological balance and overall gut health, and offering valuable insights for the development of mare milk-based functional foods.

## 1. Introduction

Mare milk, a traditional beverage among nomadic communities, is renowned for its distinct flavor and exceptional nutritional benefits. In the context of contemporary health-conscious eating and balanced diets, mare milk has garnered significant interest as a specialty dairy product. The study found that mare milk contains high concentrations of lactoferrin and immunoglobulins, which provide protective mechanisms against pathogens, support the immune system response, and enhance resistance to diseases [1]. Additionally, mare milk contains enzymes such as amylase, catalase, lipase, peroxidase, phosphatase, malate dehydrogenase, and lactoferrin, which aid in protein digestion and immune function. The digestion of proteins produces bioactive peptides with various properties, including antibacterial, anti-inflammatory, and blood pressure-regulating effects [2,3]. Kumiss, made from fermented mare milk, is known for its pleasant flavor and certain health benefits, such as aiding digestion and absorption, preventing various gastrointestinal disorders, and reducing blood lipids and cholesterol levels [4,5]. These studies highlight the significant potential of mare milk as a functional food.

Intestinal health serves as a fundamental basis for optimal immune function. As a complex microbial ecosystem, the gut microbiota plays a central role in the development and regulation of the gut immune system. Not only do they actively promote host health through their unique metabolic capabilities and increase the gut tolerance threshold [6], but the metabolites they produce also play a key role in maintaining gut homeostasis, promoting nutrient absorption, and regulating immune responses [7]. Similarly, immune dysfunction can lead to an imbalance in the gut microbiota, with an increase in harmful bacteria, thereby altering the body’s metabolic products. This triggers the release of numerous inflammatory factors, which disrupt the stability of the intestinal environment. Therefore, the gut microbiota and its metabolic activities are closely associated with the overall health of the host [8]. In recent years, interactions between food, diseases, and the gut microbiota have been reported [9]. Studies have shown that diet significantly impacts the composition of the gut microbiota, thereby profoundly influencing the health of the host [10,11]. Given the role of diet in modulating the gut microbiota-immune axis, fermented mare milk, as a potential functional food, may enhance immune function by positively altering the gut microbiota composition.

Although existing studies have shown that mare milk has immune-regulating effects, the specific mechanism underlying its relationship with the gut microbiota and immune system remains unclear, especially concerning its anti-inflammatory properties. This study hypothesizes that the anti-inflammatory effects of mare milk may be closely related to its regulation of the gut microbiota. To validate this hypothesis, the study conducted a gavage experiment with mare milk to analyze its impact on the immune indicators in mice. Additionally, the study used metagenomic sequencing technology to explore the effects of mare milk on the composition and function of the gut microbiota in mice. This research provides new insights for further investigation and development of mare milk.

## 2. Results

### 2.1. Comparison of Immune Markers

A one-way ANOVA was conducted on IFN-γ, IL-1β, IL-6, IL-10, TNF-α, and SIgA across the four sample groups (Figure 1). The IFN-γ concentration in the DW group was significantly lower than in the K and PK groups. The IL-1β concentration in the DW group was significantly lower than in the MM group but significantly higher than in both the K and PK groups. IL-6 levels were significantly elevated in the DW group compared to the MM and PK groups. The IL-10 concentration in the DW group was significantly higher than in the MM group, but significantly lower than in the PK group. TNF-α concentrations in the DW group were significantly higher than those in the K and PK groups, while SIgA levels were significantly lower than in both the MM and K groups (*p* < 0.05).

### 2.2. Analysis of Fecal Genome Characteristics

Sequencing the genomes of 32 mouse fecal samples generated a total of 204,237.1 Mb of RawBase. From this, 204,111.34 Mb of Cleanbase was obtained across all samples. Following the removal of potential host contamination, 192,849.34 Mb of Optimizedbases remained. After assembly, 3,711,524 Scaftigs were produced, with fragment sizes ranging from 500 bp to 499,597 bp (Appendix A).

For sequence alignment, DIAMOND software was utilized to match the non-redundant gene catalog sequences with those of Bacteria, Eukaryota, Viruses, and Archaea retrieved from the NR database (blastp, evalue ≤ 10^−5^). This analysis identified sequences from four kingdoms, 165 phyla, 142 classes, 253 orders, 555 families, 2144 genera, and 9618 species. At the kingdom level, Bacteria were the most abundant, followed by Eukaryota, Viruses, and Archaea. Comparisons of relative abundance at the kingdom level revealed that the PK group exhibited a higher bacterial abundance than the DW group. Additionally, the DW and MM groups showed greater Eukaryota abundance than the K group, while the DW group had a higher viral abundance compared to both the K and PK groups (Figure 2).

### 2.3. Comparison of Fecal Microbiota Among Different Groups

The relative abundance of the top 20 phyla in the fecal microbiota of mice was used to generate a heatmap (Figure 3). At the phylum level, 83 phyla were shared across all groups, while unique phyla were observed in each group: five phyla specific to the DW group, one to the MM group, three to the K group, and five to the PK group (Figure 4A). Characteristic phyla identified in the DW group included Bacteroidota, Candidatus_Gracilibacteria, and Candidatus_Wallbacteria, while Candidatus_Moranbacteria, Candidatus_Terrybacteria, and Chrysiogenota were specific to the MM group. The K group displayed characteristic phyla such as Thermoproteota, Bacillota, and Pseudomonadota, and the PK group had characteristic phyla such as Actinomycetota, Cressdnaviricota, and Nitrososphaerota (LDA > 2.5, *p* < 0.05) (Figure 4B).

At the genus level, 923 genera were common across all four groups, while 56 genera were specific to the DW group, 16 to the MM group, 32 to the K group, and 27 to the PK group (Figure 5A). The characteristic genera in the DW group included Alistipes (relative abundances of 4.80%, 2.78%, 2.42%, and 2.69% in the DW, MM, K, and PK groups, respectively), Bacteroides (3.60%, 3.21%, 2.37%, and 2.02%), Odoribacter (2.03%, 1.64%, 1.59%, and 0.99%), Duncanialla (1.71%, 1.33%, 0.94%, and 1.13%), and Prevotella (1.77%, 1.61%, 1.09%, and 1.16%), whereas Faecalibacterium (0.13%, 0.64%, 3.97%, and 2.92%) and Bifidobacterium (0.08%, 0.61%, 1.89%, and 1.68%) were characteristic of the K group. In the PK group, Enterorhabdus (0.06%, 0.34%, 0.39%, and 0.76%), Lactobacillus (1.35%, 3.67%, 4.79%, and 5.07%), and Adlercreutzia (0.4%, 1.33%, 1.54%, and 2.33%) were identified as characteristic genera (LDA > 3, *p* < 0.05) (Figure 5B).

At the species level, a total of 3179 species were shared among all four groups, with 285 species exclusive to the DW group, 97 to the MM group, 188 to the K group, and 168 to the PK group (Figure 6A). Characteristic species of the DW group included Alistipesmuris (relative abundances of 1.4%, 0.82%, 0.62%, and 0.54% in the DW, MM, K, and PK groups, respectively), Odoribacter_sp_DSM_112344 (1.27%, 0.8%, 0.71%, and 0.6%), and Muribaculaceae bacterium (5.29%, 5.25%, 3.75%, and 3.31%). The MM group exhibited characteristic species such as Erysipelotrichaceae bacterium OPF54 (0.05%, 2.75%, 1.67%, and 1.93%), Bacteroides acidifaciens (0.82%, 0.9%, 0.44%, and 0.61%), Prevotella sp. CAG 873 (0.3%, 0.31%, 0.2%, and 0.22%), and Bacteroidaceae bacterium (0.17%, 0.19%, 0.09%, and 0.11%). The K group’s characteristic species included Faecalibaculum rodentium (0.13%, 0.64%, 3.97%, and 2.92%), Bifidobacterium pseudolongum (0.07%, 0.51%, 1.4%, and 1.23%), and Limosilactobacillus reuteri (0.28%, 0.7%, 0.91%, and 0.8%). The PK group was characterized by Schaedlerella arabinosiphila (0.12%, 0.16%, 0.5%, and 0.66%), Adlercreutzia caecimuris (0.1%, 0.41%, 0.49%, and 0.82%), and Enterorhabdus sp. P55 (0.06%, 0.32%, 0.37%, and 0.72%) (LDA > 3, *p* < 0.05) (Figure 6B).

### 2.4. Functional Annotation of Fecal Microorganisms

#### 2.4.1. Functional Annotation of CAZyme

Annotation of carbohydrate metabolism functions in the gut microbiome metagenome was performed using the CAZy database. CAZy is an authoritative resource for the classification of carbohydrate-active enzymes (CAZymes), encompassing six major functional categories: glycoside hydrolases (GH), glycosyltransferases (GT), polysaccharide lyases (PL), carbohydrate esterases (CE), carbohydrate-binding modules (CBM), and auxiliary activity enzymes (AA). This annotation provides a direct reflection of the microbial functional potential to degrade complex carbohydrates. The results (Figure 7) show that among the detected CAZymes, glycoside hydrolases (GH) constitute the highest proportion, followed by glycosyltransferases (GT), carbohydrate-binding modules (CBM), and carbohydrate esterases (CE). In contrast, polysaccharide lyases (PL) and auxiliary activity enzymes (AA) represent the smallest proportions. A comparative analysis of the CAZyme profiles across the four sample groups (DW, MM, K, PK) revealed that the relative abundance of GH and CE in the DW group was significantly higher than in the MM and PK groups. The abundance of GT in the DW group was significantly lower than in the K group, while its AA abundance was significantly higher than in the K group. Furthermore, the PL abundance in the DW group was significantly higher than in both the K and PK groups. Further analysis using LEfSe (LDA > 3, Figure 8) to identify significantly differentially expressed CAZyme gene families between groups revealed that the PK group enriched four differentially expressed families, including two GT families, one GH family, and one CE family. The K group enriched 10 differentially expressed families, comprising eight GH families and two GT families. The DW group enriched eight differentially expressed families, including seven GH families and one PL family. These differentially enriched CAZyme families indicate that the different treatments (sterile distilled water, mare milk, fermented mare milk, and pasteurized fermented mare milk) had a significant impact on the functional composition of the key enzymes responsible for carbohydrate metabolism in the mouse gut microbiota.

#### 2.4.2. KEGG Functional Annotation

According to annotations from the KEGG database, the most frequently annotated genes at Level 1 were related to metabolism, while organic systems were the least represented category (Figure 8A). At Level 2, six out of the top 10 categories by relative abundance were linked to metabolic pathways (Figure 8B). The genes were annotated in the following order, from most to least abundant: carbohydrate metabolism, amino acid metabolism, metabolism of cofactors and vitamins, translation, membrane transport, energy metabolism, glycan biosynthesis and metabolism, nucleotide metabolism, replication and repair, and signal transduction.

#### 2.4.3. Cluster Analysis of Functional Relative Abundance

To analyze the core functional characteristics of the gut microbiota and the differences between groups, we conducted a functional annotation analysis using the eggNOG database. eggNOG is a widely utilized resource that categorizes genes from various organisms into orthologous groups (OGs) via unsupervised clustering and offers standardized functional classification. This annotation provides insights into the microbiota’s potential across diverse functional levels, including cellular processes, metabolism, and information storage. We selected the top 25 most abundant functions and visualized their abundance in each sample as a heatmap (Figure 9). Subsequently, clustering was performed based on functional differences. Excluding unknown functions, we found that the DW group exhibited the highest number of genes related to coenzyme transport and metabolism, the MM group had the highest number of genes related to nuclear structure, the K group showed the highest number of genes related to intracellular trafficking, secretion, and vesicular transport, and the PK group had the highest number of genes associated with chromatin structure and dynamics.

## 3. Discussion

Mare milk is rich in various bioactive compounds that contribute to immune modulation. This study found that different types of mare milk, administered via gavage, exhibited significant anti-inflammatory effects in mice, as evidenced by changes in inflammatory markers. Compared to the DW group, levels of pro-inflammatory cytokines, such as tumor necrosis factor-α (TNF-α) and interleukin-1β (IL-1β), were significantly reduced in the K and PK groups. TNF-α, a major pro-inflammatory cytokine, is secreted by immune cells (including monocytes and macrophages) and epithelial cells in tissues such as the lungs and colon [12,13,14]. As shown in Figure 1, both fermented mare milk and pasteurized fermented mare milk gavage led to a significant decrease in TNF-α levels, indicating suppression of excessive inflammatory responses. The reduction in TNF-α levels may be attributed to microbial metabolites produced during fermentation, which are known to modulate immune responses. Similarly, the decrease in IL-1β levels in the K and PK groups may result from functional peptides produced during fermentation. These peptides are believed to influence inflammatory signaling pathways, particularly by inhibiting NF-κB activity, thereby reducing IL-1β expression [15,16], a key proinflammatory cytokine. Furthermore, fermented mare milk may exert anti-inflammatory effects by activating certain subgroups of Toll-like receptors in Th2 helper lymphocytes [17,18]. This suggests that the immune-modulating properties of mare milk are multifactorial, possibly involving microbial composition and its metabolic products. In addition, the level of interleukin-10 (IL-10) in the PK group was significantly higher than in the DW group. As a key anti-inflammatory cytokine, IL-10 plays a central role in regulating the inflammatory cascade by inhibiting the activation of innate immune cells and cytokine production [19,20]. The increase in IL-10 levels in the PK group further supports the hypothesis that pasteurized fermented mare milk has potent anti-inflammatory activity, helping to downregulate the inflammatory cascade. Moreover, the IL-6 levels in the MM group were significantly lower than in the DW group, while the levels of secretory immunoglobulin A (SIgA) were markedly elevated. IL-6, a pro-inflammatory cytokine [21,22], promotes the differentiation of effector B cells and subsequent antibody production. The reduction in IL-6 levels suggests that mare milk may alleviate the inflammatory response. SIgA, a crucial antibody molecule, provides antigen-specific immune protection for the digestive tract, maintains intestinal epithelial barrier integrity, prevents pathogens from causing systemic infections, and plays a vital role in the body’s initial defense against disease [23]. The increased SIgA levels in the MM group are likely attributed to lactoferrin, a key protein found in mare milk. Previous studies have shown that lactoferrin enhances SIgA secretion by promoting the balance of Th1/Th2 cytokine profiles, further reinforcing the immune-modulating properties of mare milk [24,25,26]. Therefore, mare milk may enhance mucosal immune responses, thereby playing a regulatory role in the body’s inflammatory response. We hypothesize that the anti-inflammatory effects of mare and fermented mare milk in mice may be related to the microorganisms or their metabolites present in these milks.

The intestinal microbial community is essential in the body’s physiological functions [27]. The interactions between gut microbiota and host inflammatory metabolic pathways play a crucial role in regulating the development and progression of health and disease. On one hand, gut microbiota can produce various metabolic products, such as amino acid metabolites, which participate in modulating the host’s inflammatory response [28]. On the other hand, gut microbiota can also influence the production and release of inflammatory factors by regulating host metabolic pathway activity. This study identified Bacillota as a characteristic phylum in the K group. Bacillota ferments dietary fibers in the gut to produce metabolites such as short-chain fatty acids (e.g., butyrate and propionate), which have well-documented anti-inflammatory properties [29]. These short-chain fatty acids also promote the differentiation of regulatory T cells (Tregs), which in turn suppress excessive inflammation by secreting anti-inflammatory cytokines such as IL-10 and TGF-β [30,31]. This mechanism is consistent with the findings of this study, where IL-10 levels in Group K were higher than those in the blank control group (DW group), suggesting that fermented mare milk may exert anti-inflammatory effects through the regulation of gut microbiota. In this study, the key features identified in Group K included Faecalibaculum (relative abundances of 0.13%, 0.64%, 3.97%, and 2.92%) and Bifidobacterium (relative abundances of 0.08%, 0.61%, 1.89%, and 1.68%). Faecalibaculum derived metabolites play a significant role in these anti-inflammatory effects. They can reduce the expression of pro-inflammatory cytokines and inhibit inflammatory responses. Additionally, their metabolic products may activate the PPAR-γ signaling pathway, promote the secretion of Muc2 by intestinal epithelial cells, and enhance the expression of tight junction proteins (occludin, claudin-1), thereby improving intestinal barrier function [32]. This not only helps alleviate existing inflammation but also prevents further inflammatory damage by strengthening the intestinal barrier, ultimately achieving significant anti-inflammatory effects. Bifidobacterium, an important probiotic, is well-known for its outstanding anti-inflammatory properties. Studies have shown that Bifidobacterium can alleviate symptoms of ulcerative colitis and Crohn’s disease, reduce intestinal inflammation and damage, and demonstrate significant anti-inflammatory effects in clinical trials [33]. In a DSS (d-glucan sodium sulfate)-induced colitis mouse model, supplementation with Bifidobacterium significantly alleviated intestinal inflammation and pathological damage [34]. The mechanisms underlying these effects include the regulation of immune responses, enhancement of intestinal barrier function, and inhibition of pathogenic bacteria. These findings suggest that the anti-inflammatory properties of fermented mare milk may be closely linked to the intestinal microorganisms it contains, such as Faecalibaculum and Bifidobacterium. At the species level, the characteristic species in the MM group were Bacteroides acidifaciens (with relative abundances of 0.82%, 0.9%, 0.44%, and 0.61%) and Bacteroidaceae bacterium (with relative abundances of 0.17%, 0.19%, 0.09%, and 0.11%). The intake of mare milk increased the relative abundance of Bacteroides acidifaciens, a species known to enhance intestinal immunoglobulin A (IgA) levels, protect the intestine from pathogen infection, and alleviate inflammatory bowel disease (IBD) [35]. Additionally, the increased relative abundance of Bacteroides acidifaciens and Bacteroidaceae bacterium contributes to the restoration of gut microbiota diversity and balance, thereby alleviating DSS-induced colitis [36]. By promoting the growth of beneficial bacteria and inhibiting the proliferation of pathogenic bacteria, mare milk indirectly reduces systemic inflammatory responses and enhances the body’s anti-inflammatory capacity. In conclusion, the consumption of mare milk and fermented mare milk can regulate the composition and function of the intestinal microbiota, exerting anti-inflammatory effects at multiple levels and yielding significant anti-inflammatory outcomes.

Microorganisms and host cells are intricately interconnected, with this relationship reflected in the direct provision of substrates within the microbial food chain. In the host’s intestinal system, gut microbiota plays a key role in nutrient absorption and metabolism. This study examined the variations in dominant gut microbiota species at different taxonomic levels in mice fed various types of mare milk and provided an in-depth annotation of their metabolic pathway. Comparison of the CAZyme, KEGG, and eggNOG databases revealed that the metabolic functions of intestinal microorganisms were primarily concentrated in amino acid metabolism, carbohydrate metabolism, and the metabolism of cofactors and vitamins. These metabolic pathways may be driven by interactions among the dominant intestinal microbiota, which utilize proteins, carbohydrates, and fats as substrates to produce small molecule metabolites via fermentation and enzymolysis. These metabolites are subsequently absorbed by intestinal epithelial cells via transporters and enter the bloodstream, contributing to the body’s overall metabolic regulation. Carbohydrate metabolism plays a central role in metabolism, with the activity and stability of carbohydrate enzymes being crucial for degrading complex substrates, improving nutrient absorption efficiency in animals, and participating in essential physiological and pathological processes across various systems [37]. Carbohydrate metabolism also facilitates the fermentation of dietary fiber, leading to the production of short-chain fatty acids (SCFAs) such as butyrate and propionate [38]. SCFAs are vital in human energy metabolism and supply [39]. As microbial metabolites, they can reduce the expression of inflammatory cytokines by lowering inflammation levels, thus regulating mechanisms related to intestinal barrier function [40], microbial activity, and glucose homeostasis to maintain host health [41]. Furthermore, butyrate can directly stimulate the production of tight junction proteins in epithelial tissues, enhancing the integrity of the intestinal mucosal barrier [42]. Similarly, propionate in the gut exerts anti-inflammatory effects throughout the body [43]. As reported by Shao and colleagues, SCFAs mediate their anti-inflammatory and antioxidant effects through specific receptors such as GPR41 and GPR43 [44], which aligns with the increased IL-10 levels observed in the K/PK groups of our study. In addition, amino acid metabolism plays a crucial role in immune regulation. Amino acids are essential for immune cell function, and their availability directly influences the functional expression of immune cells, modulating the body’s immune response [45]. Tryptophan (Trp), an essential amino acid, is catabolized through complex metabolic pathways, with its metabolites serving as aryl hydrocarbon receptor (AhR) ligands. These metabolites regulate intestinal immune cell differentiation, suppress the NF-κB pathway, and downregulate pro-inflammatory cytokines such as IL-1β [46,47,48], consistent with our findings. Collectively, these results suggest that fermented mare milk may significantly impact the host’s nutrient absorption, immune regulation, and overall metabolic health by modulating the composition and metabolic functions of the gut microbiota.

## 4. Materials and Methods

### 4.1. Mare Milk Sampling

The mare milk used in this study was sourced from a farm in Tacheng, Xinjiang. During the sampling process, milk samples were collected from eight healthy mares through manual milking. Initially, the fresh milk was pooled, and a portion of the milk was added to a fermentation vessel containing residual fermented mare milk. This mixture was fermented at room temperature (20–25 °C) for 48 h. After fermentation, the fermented mare milk samples were collected. Additionally, a portion of the fermented mare milk samples was subjected to pasteurization (65 °C for 30 min) to prepare pasteurized fermented mare milk samples. All samples, including fresh mare milk, fermented mare milk, and pasteurized fermented mare milk, were stored at −20 °C for future use in subsequent experiments. The nutritional components of mare milk subjected to different treatments were presented in Table 1.

### 4.2. Animal Feeding and Sample Collection

This study involved 32 four-week-old SPF (Specific Pathogen Free) grade ICR mice (purchased from Xinjiang Medical University), with an equal number of males and females. Mice were housed in a standard animal facility under 12 h light/dark cycle with controlled humility (50 ± 5%) and temperature (22 ± 2 °C), and fed with sterilized standard chow diet and autoclaved water ad libitum. After 7 days of acclimatization, the mice were randomly divided into four groups (Table 2) (n = 8 per group, with equal numbers of males and females): sterile distilled water (DW), mare milk (MM), fermented mare milk (K), and pasteurized fermented mare milk (PK). Starting at 10:00 daily, each group received oral gavage administration of the corresponding sample (as detailed in Table 1) at a dosage of 10 mL/kg body weight, for 28 consecutive days. On day 29 of the experiment, mice were anesthetized with pentobarbital and euthanized by cervical dislocation. Liver tissue (rinsed with saline for immune index detection) and cecal content (for microbial analysis) were immediately collected and stored at −80 °C until further analysis.

### 4.3. Detection of Immune Markers

The concentrations of IL-6 (MM-0163M), SIgA (MM-0430M), TNF-α (MM-0132M), IL-1β (MM-0040M), and IFN-γ (MM-0182M) in mouse liver tissue were measured using ELISA, The kits were obtained from Jiangsu Meimian Industrial Co., Ltd. (Yancheng, China).

### 4.4. Fecal Microbial DNA Extraction and Sequencing

According to the manufacturer’s instructions, total microbial DNA was extracted from faeces using the Mag-MK Soil& Stool Genome DNA Extraction Kit (Shen gong, Shanghai, China). The concentration and quality of the extracted DNA were assessed using Qubit and agarose gel electrophoresis. Qualified DNA samples were randomly fragmented into fragments of approximately 350 bp using a Covaris ultrasonic cell disruptor, followed by end repair, A-tailing, addition of sequencing adapters, purification, PCR amplification, and other steps to complete the entire library preparation process. The quality of the prepared library is assessed using the Agilent 2100 Bioanalyzer (Agilent Technologies, Santa Clara, CA, USA) and Q-PCR. Qualified libraries are sequenced using Illumina PE150 on the Novogene Bio platform (Beijing, China).

### 4.5. Bioinformatics Analysis

The sequencing data were filtered using Readfq software (https://github.com/cjfields/readfq, accessed on 15 August 2025) to remove reads containing low-quality bases (default quality threshold ≤ 38), sequences longer than 40 bp with low-quality bases, and fragments containing more than 10 consecutive N bases, thereby generating clean data. The clean data were assembled and analyzed using MEGAHIT software (https://github.com/voutcn/megahit, accessed on 15 August 2025). Gene prediction for the assembled scaffolds was performed using MetaGeneMark (https://github.com/gatech-genemark/MetaGeneMark-2, accessed on 15 August 2025), followed by gene catalog integration and redundancy removal. Basic statistical analysis, including mean, standard deviation, and correlation analysis, was conducted on the gene abundance matrix derived from the gene catalog across the samples. Additionally, a Venn diagram was generated to visualize gene counts.

The unigenes were aligned with Bacteria, Eukaryota, Viruses, and Archaea sequences retrieved from the NCBI NR database using DIAMOND software (https://github.com/bbuchfink/diamond, accessed on 15 August 2025) (e-value threshold: 1 × 10^−5^). Taxonomic classification was performed using the Lowest Common Ancestor (LCA) algorithm. By integrating these taxonomic annotations with the gene abundance profiles, a comprehensive analysis of gene counts and relative abundance distributions across all major taxonomic ranks (kingdom, phylum, class, order, family, genus, and species) was conducted for each sample. Krona analysis was employed for visualizing phylum-level abundance, and clustering heatmaps were further used to present the results. LEfSe analysis was performed to identify key species in each group, with the LDAScore set to 2.5 at the phylum level and 3 at the genus and species levels.

For functional annotation, unigenes were aligned with functional databases using DIAMOND software (e-value threshold: 1 × 10^−5^). The functional databases used include KEGG, eggNOG, and CAZy, selected for their wide application in metagenomic functional annotation and complementary functionalities. KEGG annotations were used to analyze metabolic pathways involved in gene expression; eggNOG annotations for functional classification and clustering analysis; and CAZy annotations focused on carbohydrate-active enzymes. Based on the alignment results, statistical analysis of the relative abundance of each sample at different functional levels was performed. LEfSe analysis was employed to compare functional differences between groups, with the LDAScore threshold set to 3.

### 4.6. Data Analysis

SPSS 20.0 software was employed to conduct a one-way ANOVA on the liver immune markers and relative abundance data in mice, with a significance threshold set at *p* < 0.05.

## 5. Conclusions

This study demonstrates that mare milk significantly reduces IL-6 levels and increases SIgA levels. In contrast, fermented mare milk modulates immune responses by reducing TNF-α and IL-1β levels, while also enhancing IL-10 levels. Metagenomic analysis results indicate that mare milk promotes the proliferation of specific bacteria, such as Bacteroides acidifaciens, while fermented mare milk significantly increases the abundance of beneficial microbial communities, including Firmicutes, Enterococci, and Bifidobacteria. These changes improve the composition and function of the gut microbiota. Functional annotation analysis revealed that mare milk intervention significantly enriched key metabolic pathways, such as carbohydrate and amino acid metabolism. These metabolic activities may participate in immune modulation through the production of bioactive substances like short-chain fatty acids. In summary, this study not only provides new theoretical insights into the anti-inflammatory effects of mare milk but also offers potential reference value for its development as a functional food.

## Figures and Tables

**Figure 1 ijms-26-08239-f001:**
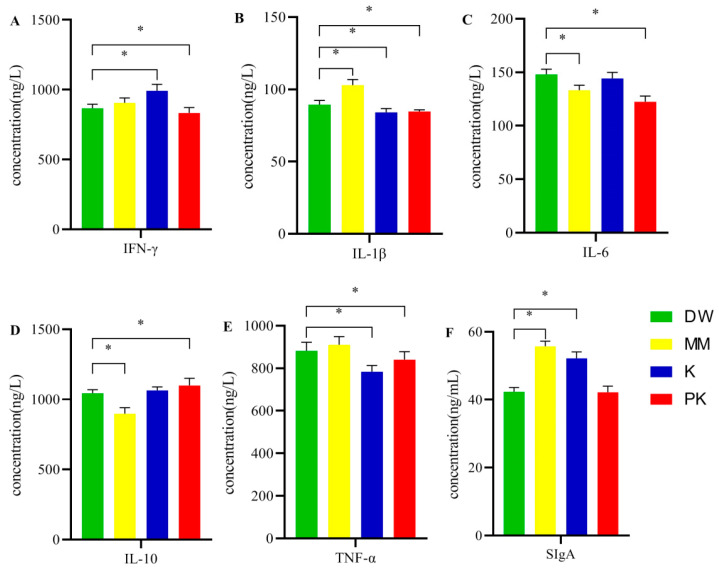
Comparison of immune markers between the sterile distilled water group (DW), mare milk group (MM), fermented mare milk group (K), and pasteurized fermented mare milk group samples. (**A**) IFN-γ concentration. (**B**) IL-1β concentration. (**C**) IL-6 concentration. (**D**) IL-10 concentration. (**E**) Concentration of TNF-α. (**F**) Concentration of SIgA. (* indicates *p* < 0.05 for comparison between two groups).

**Figure 2 ijms-26-08239-f002:**
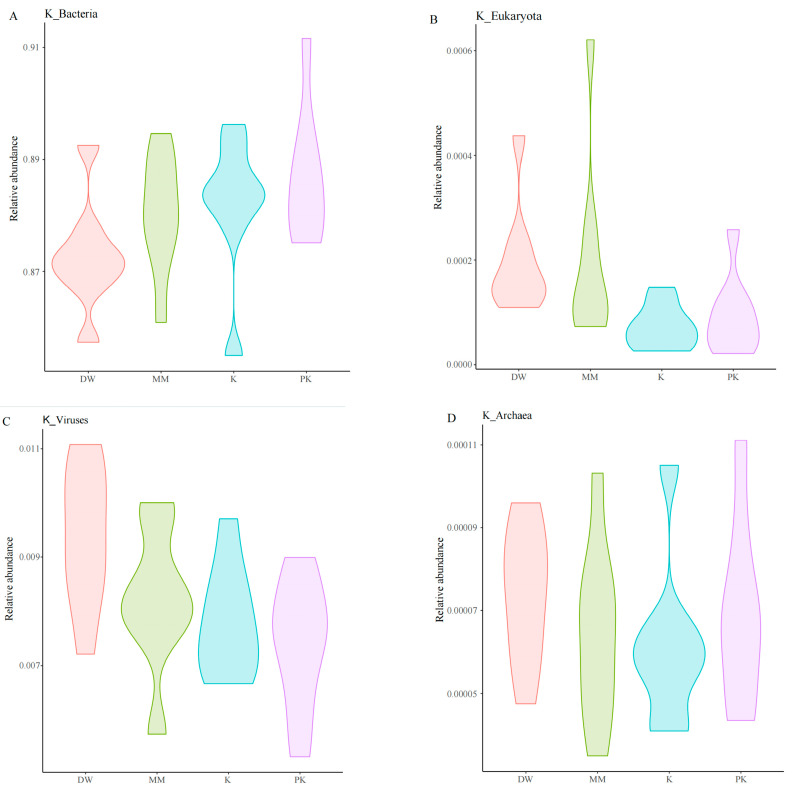
Comparison of mouse intestinal microorganisms in structural domains. (**A**) Relative abundance of four groups in Bacteria. (**B**) Relative abundance of four groups in Eukaryota. (**C**) Relative abundance of four groups in Viruses. (**D**) Relative abundance of four groups in Archaea.

**Figure 3 ijms-26-08239-f003:**
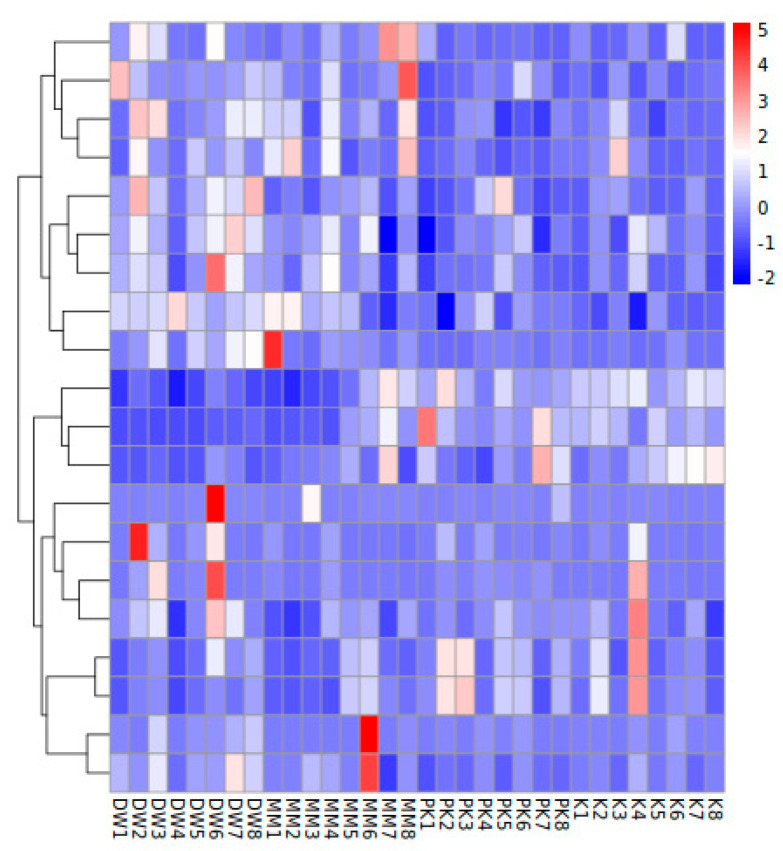
Heatmap of fecal microorganisms at phylum level.

**Figure 4 ijms-26-08239-f004:**
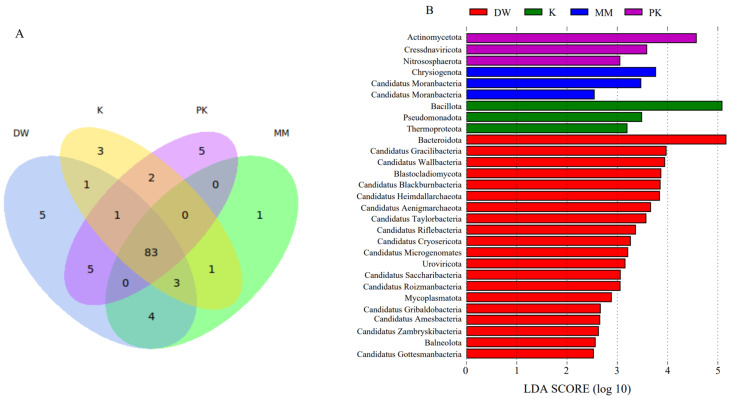
Comparison of microbial relative abundance at the phylum level among the sterile distilled water group (DW), mare milk group (MM), fermented mare milk group (K), and pasteurized fermented mare milk group (PK). (**A**) Venn diagram of different groups at the phylum level. (**B**) LEfSe analysis of different groups at the phylum level.

**Figure 5 ijms-26-08239-f005:**
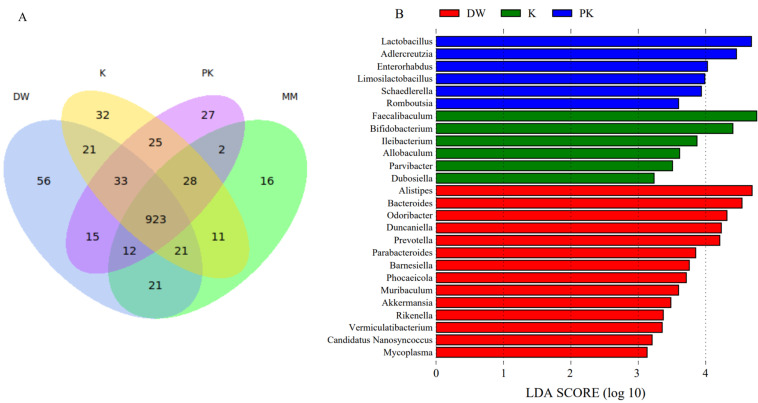
Comparison of relative microbial abundance at the genus level among the sterile distilled water group (DW), mare milk group (MM), fermented mare milk group (K), and pasteurized fermented mare milk group (PK). (**A**) Venn diagram of different groups at the genus level (**B**) LEfSe analysis of different groups at the genus level.

**Figure 6 ijms-26-08239-f006:**
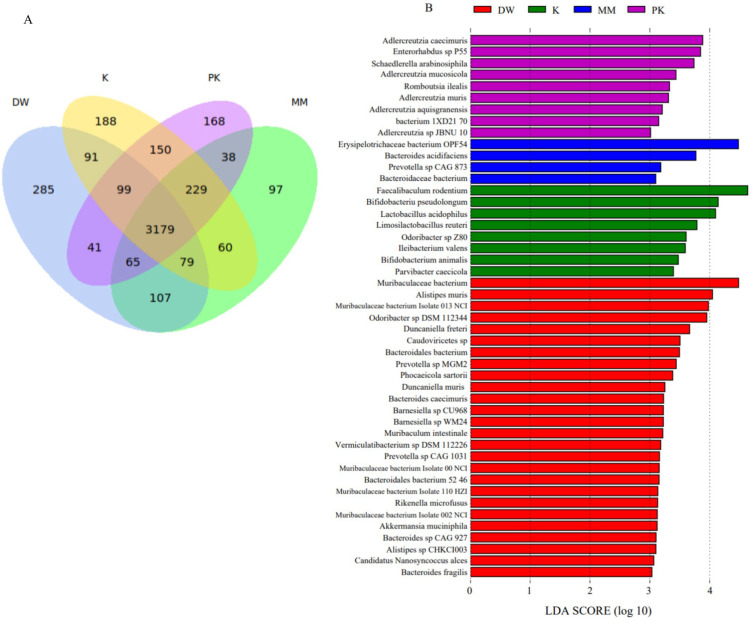
Comparison of microbial relative abundance at the species level among the sterile distilled water group (DW), mare milk group (MM), fermented mare milk group (K), and pasteurized fermented mare milk group (PK). (**A**) Venn diagram of different groups at the species level. (**B**) LEfSe analysis of different groups at the species level.

**Figure 7 ijms-26-08239-f007:**
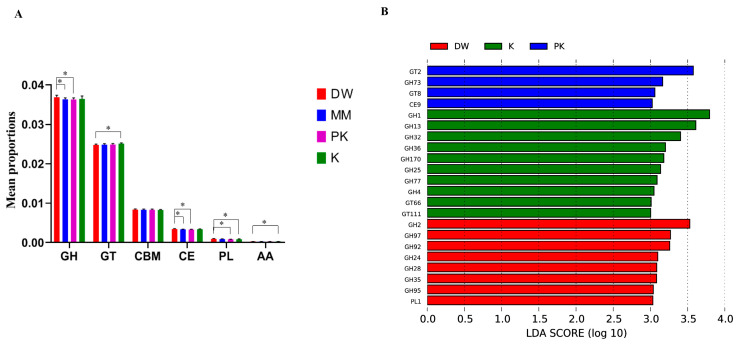
Functional annotation of mouse intestinal microbial CAZyme. (**A**)shows the comparison of annotated genes, (**B**) presents the LEfSe analysis of annotated genes. * indicates significant differences between the two groups.

**Figure 8 ijms-26-08239-f008:**
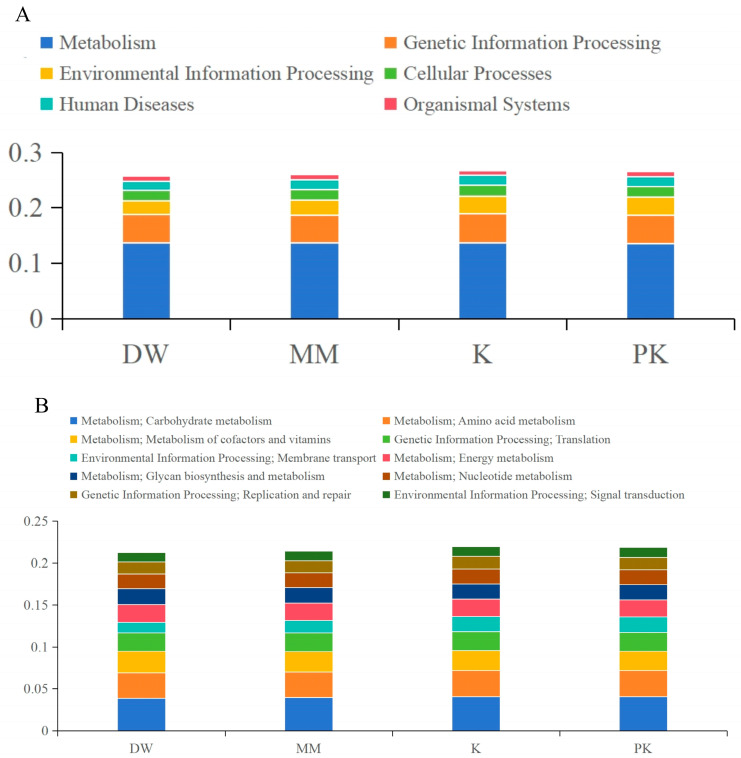
Relative abundance of functional annotations. (**A**) displays the Level 1 annotation results, (**B**) presents the Level 2 annotation results.

**Figure 9 ijms-26-08239-f009:**
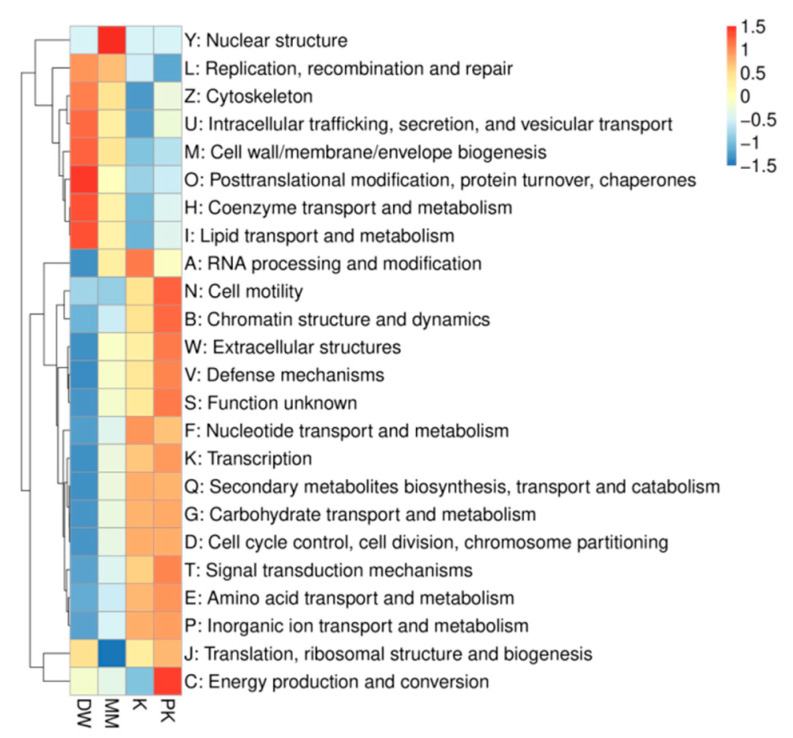
Functional abundance clustering heatmap.

**Table 1 ijms-26-08239-t001:** Nutritional components of mare milk under different treatments.

Nutritional Composition	Protein (%)	Fat (%)	Lactose (%)	Mineral (%)
mare milk (MM)	2.47	1.11	6.52	0.66
fermented mare milk (K)	2.39	1.35	1.13	0.49
pasteurized fermented mare milk (PK)	2.32	1.02	1.24	0.46

**Table 2 ijms-26-08239-t002:** Group information of the animal experiments.

Group	Administration
sterile distilled water (DW)	Sterile Distilled Water
mare milk (MM)	Mare Milk
fermented mare milk (K)	Fermented mare milk
pasteurized fermented mare milk (PK)	Pasteurized Fermented mare milk

## Data Availability

The gut microbiomics datasets presented in this study can be found in online repositories. The names of the repository/repositories and accession number(s) can be found below: Sequence Read Archive (PRJNA1237790). The original contributions presented in the study are included in the article/Appendix A, further inquiries can be directed to the corresponding author.

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
