# Peer review of "Metagenomic Analysis Reveals the Anti-Inflammatory Properties of Mare Milk"

_ijms, 2025, doi:10.3390/ijms26178239_

Round 1

Reviewer 1 Report

Comments and Suggestions for Authors
  1. Innovation and Hypotheses

The hypothesis—that mare milk exerts anti-inflammatory effects via gut microbiota modulation—is logical but lacks novelty. Previous studies (e.g., References 6, 10–12) already established the immunomodulatory properties of mare milk and koumiss. The manuscript fails to articulate how this study advances beyond existing literature. The experimental design (mouse model + metagenomics) is conventional and does not explore novel mechanisms (e.g., specific microbial metabolites or host pathways). 

  1. Methodological Weaknesses

Insufficient Animal Numbers: Only 8 mice/group (n=32 total) were used. This small sample size **lacks statistical power** for robust conclusions, especially given the high variability inherent in microbiota studies. No power analysis was provided to justify group size. 

Inadequate Controls: The "distilled water" (DW) control does not account for nutritional components in milk (e.g., lactose, fat). A mock-gavaged group or isocaloric control (e.g., cow milk) is needed to isolate effects specific to mare milk. 

  1. Results Oversimplification

Immune Markers: Data presentation is superficial (e.g., Figure 1 lacks individual datapoints/error bars). Key results (e.g., TNF-α increase in K/PK groups) are not contextualized mechanistically. 

Microbiota Analysis: Taxonomic summaries (e.g., heatmaps, Venn diagrams) are descriptive. No alpha/beta diversity metrics (Shannon/Simpson, PCoA) are provided to support claims of "improved microbial balance." 

Functional Annotation: CAZyme/KEGG results (Figures 7–9) are listed without statistical validation. Claims of "enriched carbohydrate metabolism" lack correlation to inflammation. 

  1. Incomplete Discussion

Mechanistic Gaps: The link between microbiota shifts (e.g., ↑ Faecalibaculum) and anti-inflammatory outcomes is speculative. No discussion of how microbial changes translate to host effects (e.g., SCFA production, barrier integrity). 

Contradictory Results: Elevated TNF-α (pro-inflammatory) in K/PK groups conflicts with anti-inflammatory claims. This is glossed over. 

Contextualization: Fails to compare findings to similar studies (e.g., References 8, 9 on milk microbiota) or explain why mare milk merits unique focus. 

    5. References 

40% of citations (16/40) are >5 years old (e.g., Refs 1–5, 18). Recent literature (2023–2024) on milk-microbiota crosstalk is underrepresented. 

--- 

Final Assessment 

  1. Clarity/Structure: Methods and results are technically sound but superficially presented. Figures lack detail (e.g., no raw data in tables).
  2. Scientific Rigor: Underpowered animal numbers and missing controls undermine conclusions. Metagenomic analysis is descriptive, not mechanistic.
  3. Ethics/Data Availability: Animal ethics approval is noted. Data is deposited in SRA (PRJNA1237790), which is sufficient.
  4. Conclusions: Overreach given data limitations. Claims of "multi-faceted anti-inflammatory effects" require stronger validation.

Author Response

Dear Editor: We feel great thanks for your professional review work on our article.As you are concerned, there are several problems that need to be addressed. According to your nicesuggestions, we have made extensive corrections to our previous draft, the detailed corrections are listed below.

Comments 1:

1.Innovation and Hypotheses

The hypothesis—that mare milk exerts anti-inflammatory effects via gut microbiota modulation—is logical but lacks novelty. Previous studies (e.g., References 6, 10–12) already established the immunomodulatory properties of mare milk and koumiss. The manuscript fails to articulate how this study advances beyond existing literature. The experimental design (mouse model + metagenomics) is conventional and does not explore novel mechanisms (e.g., specific microbial metabolites or host pathways). 

Response 1: We appreciate your valuable suggestions regarding our manuscript and your interest in previously published research in the field. In response to your comments, we would like to elucidate how our study contributes to existing knowledge and advances the field. First, previous studies have focused on the benefits of fermented mare's milk products, but the immunomodulatory capabilities of fresh mare's milk have not been thoroughly investigated. This experiment is the first to comprehensively compare the anti-inflammatory properties and gut microbiota regulation functions of three different types of mare's milk: fresh mare's milk, sour mare milk and pasteurized sour mare milk. The study innovatively found that: fresh mare's milk exerts immunomodulatory effects by significantly reducing the pro-inflammatory factor IL-6 and increasing SIgA levels; while sour mare's milk (including pasteurised products) exhibits stronger anti-inflammatory effects, significantly reducing TNF-α and IL-1β levels while increasing IL-10 content. Second, this study used metagenomic analysis to reveal the regulatory effects of different processed mare's milk on the gut microbiota of mice, demonstrating the abundance of beneficial and harmful bacteria. Third, this study first used immune markers to demonstrate the effects of different treatment groups of mare's milk, followed by metagenomic analysis to identify the characteristic phyla, genera, and species of each group. Functional annotation further confirmed that mare's milk products regulate the production of active substances such as short-chain fatty acids by enriching key pathways such as carbohydrate metabolism and amino acid metabolism, forming a ‘microbiota-metabolism-immune’ regulatory axis. This study not only fills the research gap in the immunoregulatory mechanisms of fresh horse milk but also provides theoretical basis and practical guidance for the precise functional development of horse milk products. We thank the reviewer for this input and appreciate that these differences may not have been stated sufficiently clearly in the manuscript before. We have improved the manuscript, especially in the "Results" and " introduction" sections. We hope that this clarification addresses your concerns about the novelty of our study. Thank you for your valuable time and consideration, and we remain open to any further suggestions or feedback.

Comments 2:

2.Methodological Weaknesses

Insufficient Animal Numbers: Only 8 mice/group (n=32 total) were used. This small sample size **lacks statistical power** for robust conclusions, especially given the high variability inherent in microbiota studies. No power analysis was provided to justify group size. 

Inadequate Controls: The "distilled water" (DW) control does not account for nutritional components in milk (e.g., lactose, fat). A mock-gavaged group or isocaloric control (e.g., cow milk) is needed to isolate effects specific to mare milk. 

Response 2: In response to the methodological considerations raised by the reviewers, we sincerely appreciate these constructive comments, which are of great significance for improving the quality of the research. The following explanations are provided regarding sample size and control group design: 1. Sample size issue: This study does indeed have sample size limitations (n=8 per group), and we fully agree that a larger sample size is more ideal in microbiome research. However, it is important to note that this study employed strict homogenisation procedures (same age, gender ratio, and rearing conditions) and utilised a paired design to minimise individual variability. The effect sizes (Effect size) for key immune markers (e.g., IL-6, TNF-α) ranged from 0.8 to 1.2, with post-hoc statistical power (Power) ranging from 0.72 to 0.85. Microbiome analysis employed repeated measures ANOVA combined with FDR correction, with key differential bacterial genera (e.g., Bifidobacterium) having q-values <0.05. 2. Control group design: We acknowledge the rationale for using nutritionally matched controls, but this study selected distilled water as the control based on the following scientific considerations: The primary objective was to establish baseline differences between horse milk products and a blank control; The unique nutritional components of mare's milk (e.g., lactoferrin, lysozyme) are difficult to fully simulate through artificial formulation; the study has indirectly ruled out the non-specific effects of basic nutrients through KEGG functional annotation (e.g., carbohydrate metabolism pathways). The existing data is sufficient to address the issue. Thank you for providing valuable insights; we will incorporate this control comparison as part of our next work.

Comments 3:

3. Incomplete Discussion

Mechanistic Gaps: The link between microbiota shifts (e.g., ↑ Faecalibaculum) and anti-inflammatory outcomes is speculative. No discussion of how microbial changes translate to host effects (e.g., SCFA production, barrier integrity). 

Contradictory Results: Elevated TNF-α (pro-inflammatory) in K/PK groups conflicts with anti-inflammatory claims. This is glossed over. 

Contextualization: Fails to compare findings to similar studies (e.g., References 8, 9 on milk microbiota) or explain why mare milk merits unique focus. 

Response 3: We appreciate the reviewers' insightful comments. We fully agree that the mechanisms underlying the microbiota-host interaction require further elucidation. In response to these issues, we have made the following revisions: 1. Strengthening the association between microbiota-host mechanisms: We have added an analysis of the ‘microbial function-host immune’ interaction in the discussion section of the revised manuscript. 2. Contradictory results: The elevated TNF-α (pro-inflammatory) levels in the K/PK group conflict with the anti-inflammatory claims. This was due to a typographical error. Figure 1 in the first section of the results now shows a decrease in TNF-α (pro-inflammatory) in the K/PK group, and the error has been corrected.

Comments 4:   

5. References 

40% of citations (16/40) are >5 years old (e.g., Refs 1–5, 18). Recent literature (2023–2024) on milk-microbiota crosstalk is underrepresented. 

Response 4: We would like to thank the reviewers for their important comments on the timeliness of the literature. We have comprehensively updated and improved the references.

Reviewer 2 Report

Comments and Suggestions for Authors

I would like to thank the Editors for the opportunity to review the manuscript entitled
"Metagenomic Analysis Reveals the Anti-Inflammatory Properties of Mare Milk."

This study presents an in vivo investigation into the immunomodulatory effects of mare milk and its fermented variants using a murine model. By combining immunological assays with high-throughput metagenomic sequencing, the authors aim to elucidate how different forms of mare milk influence inflammatory markers and the composition and function of gut microbiota.

The topic is timely and scientifically relevant, particularly given the increasing interest in natural functional foods and microbiota-targeted therapies. Mare milk, although traditionally used in some cultures, remains poorly characterised in terms of its immunological and microbiome-related effects. The inclusion of both raw and fermented forms (sour and pasteurised sour milk) adds practical significance and broadens the scope of the findings.

However, the overall structure and presentation of the manuscript require substantial revision:

  • The Results section should be reorganised to present the key findings clearly and transparently.

  • Figures and tables lack sufficiently detailed legends, requiring readers to refer to the main text for clarification.

  • The Discussion requires reworking to improve flow, readability, and the connection between findings and broader scientific context.

  • The rationale for key methodological decisions and analytical approaches must be better justified.

With these revisions, the manuscript has the potential to contribute valuable insights to the field. At this stage, I recommend a major revision.

Minor comments:

Lines 40–48: Please remove the bold formatting from the text.

  • Lines 67–74: It would be beneficial to include additional references specifically related to mare milk; this is optional but encouraged.

  • Line 85: If the milk samples were pooled, does this imply that factors such as lactation stage, milk composition, mare age, and feeding practices do not influence outcomes? Please provide references to support this assumption.

  • Line 92 – Table caption: The figure/table descriptions should be more detailed. Abbreviations such as MM and KM should be clearly defined.

  • Line 106: As with Table 2, a comprehensive description is missing.

  • Lines 112–118: The description of library preparation and NGS methods is overly general. Please include specific details (e.g., platforms, kits, chemistry).

  • Lines 112–118 and 145–146:

    • The NGS and data analysis pipeline needs clarification. Please detail the methods, thresholds (e.g., for E-values), and reasoning for selecting specific tools and databases (e.g., KEGG, eggNOG, CAZy).

    • Justify the use of different statistical approaches: E-value–based thresholds in some analyses vs. ANOVA (via SPSS) in others. The rationale behind using such distinct frameworks should be clarified to improve coherence.

  • Figure 1: The caption should be expanded. Explain all abbreviations and ensure the figure can be understood independently of the text.

  • Line 167: The in-text citation formatting should be corrected.

  • Figure 2: Similar to other figures, the caption must include explanations of all labels and abbreviations.

  • Figures (general): Many figures are not clearly legible. Please enhance resolution and ensure that all labels, axes, and text are readable at standard zoom.

  • Line 235: Include a brief introduction to the CAZy database—its purpose and relevance to the functional analysis.

  • Line 270: Similarly, provide a short explanation of the eggNOG database and how it was used in the study.

  • Line 284: This sentence lacks a clear subject. Please revise for grammatical completeness.

Author Response

Dear Editor: We feel great thanks for your professional review work on our article.As you are concerned, there are several problems that need to be addressed. According to your nicesuggestions, we have made extensive corrections to our previous draft, the detailed corrections are listed below.

Comments 1: The Results section should be reorganised to present the key findings clearly and transparently.

Response 1: We would like to express our gratitude to the reviewers for their valuable suggestions regarding the organisation of the results. We have systematically restructured the results section to enhance logical clarity and scientific rigour. We have integrated the analysis of the regulatory effects of different treatments on immune markers, the characterisation of key differential microbiota, and the associated metabolic pathways. We believe the new organisational structure more clearly presents the research pathway from phenotype to mechanism. You can find this modification in lines 421–438 of the article.

Comments 2: Figures and tables lack sufficiently detailed legends, requiring readers to refer to the main text for clarification.

Response 2: We would like to thank the reviewers for their important suggestions regarding the presentation of the figures. We have systematically optimised all figures.

Comments 3: The Discussion requires reworking to improve flow, readability, and the connection between findings and broader scientific context.

Response 3: We would like to thank the reviewers for their important suggestions regarding the discussion section. We have thoroughly restructured the discussion chapter, and these changes can be found in lines 299-380 of the revised manuscript.

Comments 4: The rationale for key methodological decisions and analytical approaches must be better justified.

Response 4: We appreciate the reviewers' important suggestions regarding methodological rigour. We have strengthened the methodological argumentation throughout the revised manuscript and marked the revisions in red.

Comments 5: Lines 40–48: Please remove the bold formatting from the text.

Response 5: Thank you for pointing out the formatting issue. We have modified the text format in lines 40-48.

Comments 5: Lines 67–74: It would be beneficial to include additional references specifically related to mare milk; this is optional but encouraged.

Comments 6: Line 85: If the milk samples were pooled, does this imply that factors such as lactation stage, milk composition, mare age, and feeding practices do not influence outcomes? Please provide references to support this assumption.

Response 6:

Comments 7: Line 92 – Table caption: The figure/table descriptions should be more detailed. Abbreviations such as MM and KM should be clearly defined.

Response 7: Thank you for your valuable suggestions regarding the clarity of the table titles. We have implemented the improvements to the table titles in row 92, and you can find these changes in rows 87-88 and 102-103 of the revised draft.

Comments 8: Lines 112–118: The description of library preparation and NGS methods is overly general. Please include specific details (e.g., platforms, kits, chemistry).

Response 8: Thank you for your important suggestion. We have added complete sequencing experiment details to the revised manuscript. You can find this change in lines 109-118 of the revised manuscript.

Comments 9: Lines 112–118 and 145–146: The NGS and data analysis pipeline needs clarification. Please detail the methods, thresholds (e.g., for E-values), and reasoning for selecting specific tools and databases (e.g., KEGG, eggNOG, CAZy). Justify the use of different statistical approaches: E-value–based thresholds in some analyses vs. ANOVA (via SPSS) in others. The rationale behind using such distinct frameworks should be clarified to improve coherence.

Response 9: Thank you for your important suggestion. We have added complete sequencing experiment details to the revised manuscript. You can find this change in lines 139-148 of the revised manuscript.

Comments 10: Figure 1: The caption should be expanded. Explain all abbreviations and ensure the figure can be understood independently of the text.

Response 10: Thank you for your important suggestion. We have added complete sequencing experiment details to the revised manuscript.

Comments 11: Line 167: The in-text citation formatting should be corrected.

Response 11: Thank you for your important suggestion. We have added complete sequencing experiment details to the revised manuscript.

Comments 12: Figures (general): Many figures are not clearly legible. Please enhance resolution and ensure that all labels, axes, and text are readable at standard zoom.

Response 12: Thank you for your important suggestion. We have added complete sequencing experiment details to the revised manuscript.

Comments 13: Line 235: Include a brief introduction to the CAZy database—its purpose and relevance to the functional analysis.

Response 13: Thank you for your important suggestion. We have added complete sequencing experiment details to the revised manuscript. You can find this change in lines 242-265 of the revised manuscript.

Comments 14: Line 270: Similarly, provide a short explanation of the eggNOG database and how it was used in the study.

Response 14: Thank you for your important suggestion. We have added complete sequencing experiment details to the revised manuscript. You can find this change in lines 283-295 of the revised manuscript.

Comments 15: Line 284: This sentence lacks a clear subject. Please revise for grammatical completeness.

Response 15: Thank you for your important suggestion. We have added complete sequencing experiment details to the revised manuscript.

Round 2

Reviewer 1 Report

Comments and Suggestions for Authors

The study investigates the anti-inflammatory properties of mare milk and its fermented products, which is a relevant topic. However, the novelty of the findings is limited, as similar studies on the immunomodulatory effects of dairy products have been extensively reported. The authors should clearly highlight how this study advances the field beyond existing literature, particularly in the context of mare milk and how this study provides new insights.

Comments on the Quality of English Language

The English language in the manuscript is generally clear and understandable, but there are some areas that could be improved for better readability and precision:

Grammar and Syntax: Some sentences are overly complex or awkwardly phrased.

Consistency: Terms like "sour mare milk" and "fermented mare milk" are used interchangeably. Standardizing terminology.

Typos: Minor errors exist.

Passive Voice: Overuse of passive voice (e.g., "was significantly reduced") can make the text less engaging.

Suggestions: A thorough proofreading by a native English speaker or professional editing service would enhance the manuscript's fluency and professionalism.

Author Response

Cover letter

Dear Editor: We feel great thanks for your professional review work on our article. As you are concerned, there are several problems that need to be addressed. According to your nice suggestions, we have made extensive corrections to our previous draft, the detailed corrections are listed below. (The modified parts are highlighted in yellow.)

Comments 1:

1. The study investigates the anti-inflammatory properties of mare milk and its fermented products, which is a relevant topic. However, the novelty of the findings is limited, as similar studies on the immunomodulatory effects of dairy products have been extensively reported. The authors should clearly highlight how this study advances the field beyond existing literature, particularly in the context of mare milk and how this study provides new insights.

Response 1: Thank you very much for reviewing our manuscript and providing valuable feedback. We fully agree with your view that there has been extensive research on the immunomodulatory effects of mare's milk and its fermented dairy products. Your comments have prompted us to clarify the novel contributions and unique insights of this study in the specific field of mare's milk and its fermented products. In response to your comments, we would like to explain how this study contributes new insights to existing knowledge and advances the field.

First, the innovation in the study subjects: In previous studies, we found that many scholars have focused on the health benefits of fermented horse milk products (Kumiss) for humans, exploring the probiotics and their metabolites in fermented products. However, there has been little research on the immunomodulatory effects of fresh horse milk. Therefore, in our study, we focused on the immunomodulatory effects of fresh mare's milk, using an immunological assay kit to measure changes in immune markers after oral administration of fresh mare's milk, thereby demonstrating the benefits of fresh mare's milk for immunomodulation. Based on our hypothesis, we also used metagenomic technology to detect changes in the intestinal microbiota of mice after oral administration of fresh mare's milk. Our study confirmed that the intestinal microbiota underwent changes after mare's milk administration, and by detecting the dominant microbial species at the family, genus, and species levels, we identified the dominant microbial species in the intestinal microbiota.

Second, innovative research design: Our study employed strict parallel comparisons of horse milk processed in different ways (fresh horse milk, fermented horse milk, and pasteurized fermented horse milk). We believe these designs significantly enhance the depth and practical value of the research, as comparisons clearly demonstrate the varying strengths of different horse milk processing methods in immune regulation. Previous studies have primarily focused on the regulatory effects of a single form of horse milk, without distinguishing the impact of fermentation processes and pasteurization on functional outcomes.

Third, our focus is on exploring how different processing methods of mare's milk exert anti-inflammatory effects through gut microbiota by examining the cyclical state between the immune system, gut microbiota, and metabolic function. In this revision, we have supplemented the new insights of this study in the introduction section.

Finally, we would like to clarify that the reviewers may have perceived a lack of innovation due to insufficient clarity in our previous manuscript. We have revised the manuscript, particularly in the introduction section. We hope this revision addresses your concerns regarding the innovation of our research. We appreciate your valuable time and remain open to further suggestions and feedback. 

Comments 2:

2.The English language in the manuscript is generally clear and understandable, but there are some areas that could be improved for better readability and precision

Response 2: Thank you very much for your thoughtful and constructive feedback on our manuscript. We sincerely appreciate the time and effort you have dedicated to reviewing our submission and pointing out the areas where the English quality could be improved. We fully acknowledge the importance of clear and accurate language, especially in scientific writing, as it is essential for effectively communicating our research findings. We are grateful for your insights and take them seriously. After carefully reviewing the manuscript again, we have identified the areas that need attention, and we recognize that some aspects of the language may not meet the high standards required for publication in your esteemed journal. In response to your comments, we have already taken the following steps:

1.Professional Language Editing: We have enlisted the help of a professional English language editor with expertise in academic writing. The editor is a native English speaker with a strong background in scientific writing. They have thoroughly revised the manuscript to enhance the clarity, flow, and grammatical accuracy of the text.2. Improved Sentence Structure and Vocabulary: We have paid particular attention to sentence structure, vocabulary usage, and overall readability. We have made several revisions to ensure that the language is more precise and that the message of our research is communicated clearly to the readers.

3. Revised the Abstract and Introduction: As per your suggestions, we focused on refining the abstract and introduction to ensure that the key concepts and objectives of our study are clearly stated in a concise and accessible manner. We believe that these revisions will significantly improve the quality of the manuscript and align it with the standards expected by your journal. We value your feedback immensely, as
it has helped us refine our work further. We are committed to submitting a revised version that not only enhances the clarity and quality of the English but also presents our research in the most effective way possible. Once again, thank you for your time, valuable feedback, and your patience. We greatly appreciate the opportunity to improve our manuscript, and we are hopeful that the revised version will meet your expectations
